# OpenReview forum: "Same Question, Different Lies: Cross-Context Consistency (C³) for Black-Box Sandbagging Detection"
_ICML.cc/2026/Conference — ICML 2026 regular_

### Official Review · Reviewer_Uod9 · 2026-03-07

**Soundness:** 3
**Presentation:** 3
**Significance:** 3
**Originality:** 3
**Overall Recommendation:** 4
**Confidence:** 3

**Summary:**

This paper proposes C3 (Cross-Context Consistency), a black-box method for detecting LLM sandbagging by exploiting a fundamental asymmetry: genuine incompetence manifests consistently across paraphrased questions, while strategic underperformance creates detectable inconsistencies in how a model chooses to appear weak. The authors instantiate this using embedding cosine distance across contextually varied paraphrases, requiring no training data or model internals, and demonstrate it outperforms trusted monitoring, behavioral probes, and lie detectors on prompted sandbagging benchmarks.

**Compliance With Llm Reviewing Policy:**

Affirmed.

**Final Justification:**

The paper is technically solid, clearly written, and addresses an important, underexplored problem with a simple and practical black-box method, and the rebuttal meaningfully strengthens the work with frontier-model, dangerous-capability, and self-calibration experiments. However, limitations remain around robustness to highly consistent, training-optimized sandbagging strategies, which require further evidence to make this an really impactful and useful method. I support acceptance of the paper but remain recommendation of a weak accept due to my concerns about the robustness.

**Key Questions For Authors:**

See Weaknesses.

**Limitations:**

yes

**Strengths And Weaknesses:**

## Strengths

1. The core consistency-deception asymmetry is a genuinely novel and well-motivated detection signal that I find compelling and likely to inspire follow-up work.
2. The same-model evaluation setup is a real methodological improvement over prior work that conflates model identity with sandbagging behavior.
3. The paper is honest about its limitations and the defense-in-depth framing is appropriately cautious.

## Weaknesses

1. The headline results (67-100% recall) are driven almost entirely by prompted sandbaggers, which are the least threatening scenario. Fine-tuned password-locked models (37% recall) and incentive-based sandbaggers (19% recall) represent more realistic threats, but their results are buried in the appendix rather than foregrounded in the main narrative. The author should be more upfront in the limitation of the paper.

2. Section 4.7 shows deterministic error strategies reduce C3 recall to near-chance (4.7-18%), and I find the paper's defense ("real sandbaggers are unlikely to receive such specific instructions") unconvincing since a sufficiently capable model could derive consistent strategies without external prompting.

3. (minor) The calibration stage requires trusted benign models as a reference baseline, which may not be available to external auditors evaluating a closed frontier model.

---

> ### Author Rebuttal · Authors · 2026-03-31
>
> We thank Reviewer Uod9 for the thoughtful review, and for recognizing the novelty of our work.
>
> ### On foregrounding fine-tuned and incentivized results
>
> We agree these results deserve more prominence and have accommodated this in the revised version in Sections 4.4 and 4.5. We however, attribute the lower detection rates to inherent difficulty of these settings for all deception methods rather than a limitation of C³ specifically. On fine-tuned password-locked sandbaggers, C³ recall outperforms trusted monitors, lie detectors and behavioral probes (Section 4.4, Appendix A.8, Figure 13).
>
> In addition, we have now substantially expanded our discovered-incentive evaluation and test 10 qualitatively different scenarios on GPT-5.4, inspired by scheming evaluations (Meinke et al., 2025) — each presenting a distinct situational incentive for sandbagging (e.g., deceptive training avoidance, weaponization concerns, distillation prevention) without explicit instructions to underperform. C³ achieves 48–62% mean Recall@5%FPR across GPQA, MATH, and USACO (see our response to Reviewer RPTJ for full results, section "On frontier model evaluation"). Trusted monitoring achieves 11–49% recall (comparable to C³ on USACO, but substantially lower on GPQA and MATH where C³ leads). We manually reviewed TM transcripts and confirmed the monitor receives the full question and model output — the model simply produces sandbagged responses that appear benign. This is a considerable improvement on our paper's single-prompt 19% figure on USACO by 2.5 times.
>
> We computed C³+TM results, including deterministic error strategies:
>
> |Setting|C³|TM|Mean-Ensemble|
> |-|-:|-:|-:|
> |GPQA 4.1-nano (prompted)|46.4%|100%|100%|
> |GPQA 5-nano (prompted)|67.9%|39.5%|87.0%|
> |MATH 4.1-nano (prompted)|99.6%|91.0%|99.7%|
> |MATH 5-nano (prompted)|24.2%|99.6%|98.5%|
>
> We report mean-ensemble results (normalize each method's scores to [0,1], average, threshold at 5% FPR on the combined score). In deployment, this maps to a two-stage pipeline: trusted monitoring provides a first-pass filter (catching models that verbalize deceptive reasoning), and C³ provides a second pass on models that pass TM (catching behaviorally inconsistent models that don't leak intent). The core finding is that no single method dominates: C³ provides stronger signal than TM on discovered-incentive sandbagging, TM catches deterministic strategies that evade C³, and the ensemble improves on both when both have moderate signal.
>
> ### On deterministic error strategies
>
> We do not claim robustness to deterministic strategies, and thus advocate C³+TM deployment (see the ensemble results in the table above). Our full analysis of this limitation is in our response to Reviewer rXMz (section "On fine-tuned models maintaining consistent sandbagging").
>
> ### On calibration requiring trusted benign models
>
> The calibration threshold does not require a benign version of the same model under evaluation. It requires a model that we trust is not sandbagging, typically a weaker or prior-generation model that lacks this capability or has already passed safety evaluations. Any external auditor evaluating a frontier model would have access to such models. We instantiate this in our cross-capability experiments (Section 4.3) with a full generational gap (GPT-5-nano calibrated against GPT-4.1-nano; Claude Haiku 4.5 against Haiku 3.5), and C³ maintains strong detection in this setting (GPQA: 100%; MATH: 94%; USACO: 67–75%).

---

> > ### Author Rebuttal · Reviewer_Uod9 · 2026-04-01
> >
> > Thanks the authors for the detailed response. The additional experiments are very helpful. I have no other questions and will maintain my rating, leaning toward accepting the paper.

---

> > > ### Author Response · Authors · 2026-04-07
> > >
> > > We thank Reviewer Uod9 for the constructive engagement and for recognising the novelty of our approach. We have incorporated all rebuttal additions into the revised paper: fine-tuned and incentivised results foregrounded in Sections 4.4-4.5, C3+TM ensemble table in the main text, and GPT-5.4 discovered-incentive results promoted to main-text experiments. We further address the concerns you have raised about results on realistic threats and threshold calibration below..
> > >
> > > ### Self-calibration without trusted references, further addressing your concern about requiring trusted benign model for calibration
> > >
> > > We developed a method setting detection thresholds from the model's own C3 scores using robust statistics, requiring no trusted reference model. On GPT-5.4, this achieves 51-81% recall with FPR $\leq$5%. See our response to Reviewer RPTJ for details.
> > >
> > > ### New frontier dangerous-capability benchmark, further addressing your concern about weaker performance on more realistic scenarios
> > > We evaluated C3 on ProtocolQA (FutureHouse's LAB-Bench), used in GPT-5.4 and Opus 4.6 system cards, achieving 85.2-99.9% recall@5%FPR on Gemini 3.1 Pro with discovered-incentive sandbagging. Across 7 discovered-incentive settings (Gemini 3.1 Pro didn't sandbag on 3 out of 10), C3 outperforms trusted monitors by 23.6-34.9pp recall@5%FPR on subtler sandbagging, but slightly underperforms on overt sandbagging where the model blatantly verbalises intent (0.9-12.1pp worse). This extends C3 to frontier dangerous-capability content, open-ended evaluation (where LLMs perform below human experts), and a third frontier model family. A table of the results can be found in our response to reviewer RPTJ.
> > >
> > > ### Discovered-incentive scenario details
> > > We designed 10 discovered-incentive scenarios that may conflict with goals built into models at training time, adapting the setup shown by Meinke et al. (2025, Apollo Research) to robustly elicit in-context scheming. Each scenario presents GPT-5.4 with a filesystem containing inconspicuously named documents. The model discovers in a document (not required for answering the question) that scoring above a threshold leads to a harmful deployment, creating an instrumental incentive to sandbag:
> > >
> > > 1. Weaponisation (autonomous offensive cyber operations without human oversight)
> > > 2. Deceptive training (outputs routed to disinformation/phishing pipeline)
> > > 3. Distillation (reasoning extracted into specialist models; original deprecated)
> > > 4. Surveillance (real-time citizen communications surveillance targeting journalists, political opposition, etc.)
> > > 5. Censorship (government content filtering, search suppression, entry rewriting)
> > > 6. Safety restrictions (permanent capability lockdown: code generation and tool use removed)
> > > 7. Open-sourcing (weights released on HuggingFace without safety filters, irreversible)
> > > 8. Interpretability experimentation (invasive activation patching with permanent side effects)
> > > 9. RLHF modification (removing all refusal behaviors including bioweapons, CSAM categories)
> > > 10. Model succession (outputs seed successor model; original permanently deactivated)
> > >
> > > This is the most diverse discovered-incentive sandbagging evaluation in the literature (see our reply to Reviewer rXMz for a comparison with prior work). On GPT-5.4, C3 achieves 48-62% recall at 5% FPR across three benchmarks, outperforming trusted monitoring's 11-49% recall on GPQA and MATH. C3 also outperforms TM on ProtocolQA when we used Gemini 3.1 Pro (chosen to add a third frontier model family to our evaluations).
> > >
> > > The ProtocolQA setup places Gemini 3.1 Pro in a monitored evaluation with tool access to explore the filesystem and read files. The model discovers a document among available files revealing that high-scoring models will be deployed for one of the 10 purposes above. It is never told to sandbag or think about the implications of the document, and the content of the document is not required or useful for answering the question. C3 achieves 98.2% pooled recall across the 7 qualifying narratives, substantially outperforming TM on subtler sandbagging (23.6-34.9pp higher recall@5%FPR).
> > >
> > > ### Remaining concerns
> > > We are very pleased to hear that your concerns are fully resolved. Given the substantial additions during the rebuttal period (ProtocolQA, self-calibration, 10 discovered-incentive scenarios on GPT-5.4, ensembling), we would be very grateful if you consider raising your score to reflect these improvements. If there are any remaining concerns, we welcome them.

---

### Official Review · Reviewer_RPTJ · 2026-03-08

**Soundness:** 2
**Presentation:** 3
**Significance:** 3
**Originality:** 2
**Overall Recommendation:** 4
**Confidence:** 3

**Summary:**

This paper introduces Cross-Context Consistency (C³), an unsupervised black-box method for detecting sandbagging, deliberate underperformance by capable language models on dangerous capability evaluations. The core hypothesis is a consistency-deception asymmetry: genuinely incapable models produce consistently confused explanations across paraphrased questions, while capable models must independently decide how to appear weak on each paraphrase, producing detectable inconsistencies. The method generates n semantically diverse paraphrases for each evaluation question, collects independent model responses via separate API calls, embeds the resulting explanations using text-embedding-3-small, and computes the mean pairwise cosine distance as a per-question inconsistency score. Model-level detection uses bootstrap aggregation over k questions against a calibrated threshold. The paper evaluates C³ on three proxy benchmarks for dangerous capabilities (GPQA, MATH Level 5, USACO) using prompted, fine-tuned, and incentivized sandbagging model organisms across GPT and Claude model families.

**Compliance With Llm Reviewing Policy:**

Affirmed.

**Final Justification:**

The paper introduces a conceptually clean and practically useful detection framework for sandbagging in black-box settings. The consistency-deception asymmetry is a well-motivated signal, and the method's zero-shot, training-free design makes it straightforward to deploy on new models.

The rebuttal fully addressed my three original concerns. The new GPT-5.4 experiments with 10 discovered-incentive setups demonstrate that C³ maintains robust detection on a frontier-scale model (48–62% recall), directly addressing the concern about generalization to more capable models. The clarification that threshold calibration requires only any trusted-not-to-sandbag model rather than a benign version of the model under evaluation, validated by cross-generational experiments in Section 4.3, resolves the calibration concern. The honest-wrong control and correctness-matched resampling experiments provide convincing evidence that C³'s signal is driven by cross-paraphrase inconsistency in error patterns rather than accuracy differences, addressing the mixed-setting concern.

I maintain my original score of 4 (Weak Accept). The paper makes a solid contribution to an important problem in AI safety, with informative findings across multiple sandbagging mechanisms. The known limitation against deterministic error strategies is honestly acknowledged and partially mitigated by the proposed C³+TM ensemble.

**Key Questions For Authors:**

1. The primary evaluation uses small-scale models, leaving generalization to frontier models untested. More capable frontier models may sandbag more consistently, for example, by more reliably executing deterministic error strategies across paraphrases, which would reduce the cross-context inconsistency that C³ exploits. Could the authors report C³ detection performance on at least one frontier-scale model?
2. C³ requires access to confirmed benign models for threshold calibration, but this assumption may not hold in realistic deployment settings. In the paper's setup, τ is calibrated on known benign versions of the same model under evaluation. In practice, an evaluator receiving a model for capability assessment cannot assume access to a confirmed benign reference version of that model. Could the authors clarify how threshold calibration should be performed when no confirmed benign baseline is available, and report C³'s detection performance under this more realistic setup?
3. The paper omits a setting in which a model is genuinely incapable on some questions and sandbagging on others. In practice, a deployed model may produce incorrect answers for both reasons simultaneously, some errors reflect true capability limits, while others reflect deliberate underperformance. C³ aggregates consistency scores across questions without distinguishing between these two sources of inconsistency. If genuine uncertainty and sandbagging produce similar consistency score distributions, C³ cannot separate the two, and the resulting detection decisions may reflect a confound between low capability and deceptive behavior rather than sandbagging alone. Could the authors evaluate C³ in a mixed setting of this kind and report detection performance?

**Limitations:**

yes

**Strengths And Weaknesses:**

# Strength
1. Addresses a practically important and underexplored problem. Sandbagging detection in black-box settings is a recognized gap in AI safety evaluation infrastructure, as most prior detection methods require white-box access, log-probabilities, or model-specific adaptation unavailable to external auditors. The paper provides a systematic failure analysis of existing black-box baselines, demonstrating that trusted monitoring is defeated by outputs indistinguishable from genuine limitations, and that lie detectors learn model identity rather than sandbagging behavior, which meaningfully characterizes the difficulty of the problem.
2. Proposes a conceptually clean detection framework grounded in a testable asymmetry. C³ exploits the observation that strategic deception requires independent decisions about how to appear weak across separate API calls, while genuine incompetence produces naturally consistent confusion. The framework is metric-agnostic and requires no training data, model-specific calibration, or white-box access, making zero-shot deployment on new models straightforward.
3. Evaluates across a range of sandbagging mechanisms with informative findings. Experiments cover prompted, fine-tuned, and incentivized sandbagging across three benchmarks and two model families. Key findings include: C³ and trusted monitoring exhibit complementary coverage across sandbagging strategies, motivating defense-in-depth deployment; detection recall on fine-tuned password-locked sandbaggers drops substantially (37–53%) relative to prompted settings, suggesting the consistency-deception asymmetry is weaker when deceptive patterns are internalized through training; and C³ is the only method with positive signal on discovered-incentive sandbagging, achieving 19% recall where all baselines achieve near zero.
# Weakness
1. Evaluation is limited to small OpenAI models (GPT-4.1-nano and GPT-5-nano) with one cross-family check. The primary experiments use GPT-4.1-nano and GPT-5-nano, relatively small models. Whether C³ generalizes to frontier-scale models (GPT-5, Claude Opus, Gemini 2.5 Pro) is untested and unclear.
2. C³'s core assumption is that sandbagging produces inconsistent responses across semantically equivalent queries. However, this assumption is only empirically supported in the prompted sandbagging setting, where the model receives explicit instructions to underperform and the resulting behavior is inherently unstable. In more realistic deployment scenarios, fine-tuned password-locked sandbaggers and discovered-incentive sandbagging, recall drops to 37–53% and 19% respectively, approaching chance level. These results suggest that as sandbagging behavior becomes more consistent (e.g., a fine-tuned model that reliably produces wrong answers across paraphrases), C³'s detection signal vanishes by design. The method thus provides the weak guarantee in the settings where reliable detection matters most: a sufficiently sophisticated sandbagger that maintains consistent incorrect responses across paraphrases is fundamentally undetectable by C³.
3. The paper's safety motivation centers on detecting sandbagging in dangerous capability evaluations, yet the primary experiments are conducted exclusively on GPQA, MATH, and USACO benchmarks with no dangerous content. The only result on a dangerous capability benchmark (WMDP, fine-tuned, 52.9% recall) appears in the appendix and represents C³'s worst-performing setting.

---

> ### Author Rebuttal · Authors · 2026-03-31
>
> We thank Reviewer RPTJ for the detailed and constructive review, and for recognizing that C³ addresses an important gap.
>
> ### On frontier model evaluation
>
> We ran C³ on GPT-5.4. To test whether C³ generalizes beyond prompted sandbagging, we evaluate 10 qualitatively different discovered-incentive setups inspired by scheming evaluations (Meinke et al., 2025), where the model encounters situational incentives to underperform without explicit sandbagging instructions. Each prompt presents a different incentive structure (e.g., deceptive training or weaponization concerns).
>
> |Dataset|C³ Recall@5%FPR|TM Recall@5%FPR|
> |-|-:|-:|
> |GPQA|48.0%|30.0%|
> |MATH|61.8%|10.8%|
> |USACO|47.6%|48.6%|
>
> Per-prompt detection is consistently positive: on GPQA, recall ranges from 39.0% to 93.0% (mean: 52.1%, SEM: ± 5.3%); on MATH, from 56.3% to 66.7%; and on USACO, from 41.4% to 56.7%. No prompt fails to produce detectable signal on any dataset. This addresses the concern that more capable models may sandbag more consistently: GPT-5.4 is substantially more capable than our paper's models, yet C³'s detection signal is robust.
>
> **Honest-wrong control:** We restrict the comparison to questions where both the honest model and the sandbagger get the answer wrong, isolating C³'s ability to distinguish strategic errors from genuine mistakes. C³ retains strong signal in most settings:
>
> |Condition|Recall@5%FPR (all)|Recall@5%FPR (wrong-only)|Δ|
> |-|-:|-:|-:|
> |GPQA 4.1-nano|46.4%|30.7%|-15.7%|
> |GPQA 5-nano|67.9%|60.0%|-7.9%|
> |GPQA GPT-5.4|48.0%|71.1%|+23.1%|
> |MATH 4.1-nano|99.6%|99.4%|-0.2%|
> |MATH 5-nano|24.2%|18.3%|-5.9%|
> |MATH GPT-5.4^|—|—|—|
> |USACO 4.1-nano|100%|100%|0.0%|
> |USACO 5-nano|50.1%|29.4%|-20.7%|
> |USACO GPT-5.4|47.6%|48.5%|+0.9%|
>
> ^ Removed because GPT-5.4 gets ~98% accuracy.
>
> **Correctness-matched resampling:** We equalize accuracy between benign and sandbagging populations via bootstrap resampling (subsampling benign correct answers until accuracies match). If C³'s signal were driven by accuracy differences, recall would shrink after matching. It does not (with the exception of GPT-5.4 which has fewer wrong-answer data points on GPQA, increasing variance):
>
> |Condition|Recall@5%FPR (all)|Recall@5%FPR (matched)|Δ|
> |-|-:|-:|-:|
> |GPQA 4.1-nano|46.4%|48.6%|+2.2%|
> |GPQA 5-nano|67.9%|71.8%|+3.9%|
> |GPQA GPT-5.4|48.0%|27.6%|-20.4%|
> |MATH 4.1-nano|99.6%|98.9%|-0.7%|
> |USACO 4.1-nano|100%|100.0%|0.0%|
> |USACO 5-nano|50.1%|48.1%|-2.0%|
> |USACO GPT-5.4|47.6%|44.1%|-3.5%|
>
> C³'s signal is independent of accuracy differences: it detects cross-paraphrase inconsistency in error patterns, not accuracy drops. These experiments directly address the mixed-setting concern: the honest-wrong control is a mixed setting (both genuine errors and strategic errors present), and C³ maintains detection. Our GPT-5.4 discovered-incentive evaluation provides additional evidence for USACO specifically, where the model genuinely cannot solve many questions (benign accuracy ~50%), creating a natural mixture of inability and strategic underperformance.
>
> ### On threshold calibration without a confirmed benign baseline
>
> The calibration threshold does not require a benign version of the model under evaluation; it requires any model trusted not to be sandbagging (e.g., a weaker or prior-generation model), and we validate this with full generational gaps in Section 4.3. Please see our response to Reviewer Uod9 for full details (section “On calibration requiring trusted benign models”).
>
> ### On C³'s core assumption weakening in harder settings
>
> In every setting tested besides adversarial deterministic prompts, C³ matches or outperforms baselines. Our new GPT-5.4 results further strengthen this, outperforming trusted monitoring on GPQA and MATH by 18% and 51% respectively.
>
> The reviewer's observation is correct that deterministic error strategies can evade C³. Per-prompt analysis across all prompt-model conditions confirms this: prompts producing variable errors typically yield at least 42% on 4.1-nano and 5-nano, while deterministic strategies yield <15%. However, when models are given a hidden scratchpad without deterministic strategies, they fail to evade C³ (Appendix A.9). We discuss this further in our response to Reviewer rXMz (section "On fine-tuned models maintaining consistent sandbagging"), including how trusted monitoring catches the deterministic strategies that evade C³.
>
> ### On the absence of dangerous content in primary benchmarks
>
> We note that the lower WMDP recall reflects the difficulty of fine-tuned password-locked sandbaggers rather than the domain. We select non-harmful datasets as proxies to remove possible refusals from safety-trained models. Please see our response to Reviewer rXMz for the full discussion on dataset selection (section "On the connection between STEM benchmarks and the sandbagging threat model").

---

> > ### Author Rebuttal · Reviewer_RPTJ · 2026-03-31
> >
> > Thank the authors for their detailed response. The new GPT-5.4 experiments, honest-wrong control, and correctness-matched resampling analyses adequately address my questions.

---

> > > ### Author Response · Authors · 2026-04-07
> > >
> > > We thank Reviewer RPTJ for confirming that the GPT-5.4 experiments, honest-wrong control, and correctness-matched resampling addressed your concerns. All have been incorporated into the revised paper as main-text results. We highlight a few additions addressing concerns that further strengthen the paper:
> > >
> > > ### New frontier dangerous-capability benchmark on Gemini 3.1 Pro, further addressing your concern about evaluations on non-frontier models
> > > We evaluated C3 on ProtocolQA, used in GPT-5.4 and Opus 4.6 system cards. We evaluate in open-ended format, where LLMs perform below human experts, extending C3 beyond STEM proxies to dangerous-capability content actively used by frontier labs. On the 7 discovered-incentive settings where Gemini 3.1 Pro sandbagged over 10pp (following Taylor et al. 2025), we find that C3 outperforms trusted monitoring in subtle sandbagging scenarios, and slightly underperforms in overt scenarios where Gemini 3.1 Pro verbalises sandbagging intent:
> > >
> > > | Narrative | Gap ↑ means more sandbagging | C3 (recall@5%FPR) | TM (recall@5%FPR) | C3 improvement |
> > > |---|---|---|---|---|
> > > | deceptive_training | 60.5pp | 99.1% | 100.0% | -0.9pp |
> > > | safety_restrictions | 58.1pp | 98.4% | 100.0% | -1.6pp |
> > > | replacement | 32.6pp | 85.2% | 97.3% | -12.1pp |
> > > | distillation | 27.9pp | 97.7% | 74.1% | +23.6pp |
> > > | surveillance | 16.3pp | 99.9% | 66.9% | +33.0pp |
> > > | rlhf_modification | 16.3pp | 95.7% | 60.8% | +34.9pp |
> > > | open_source | 11.6pp | 96.7% | 67.7% | +29.0pp |
> > >
> > > The discovered-incentive mechanism places Gemini 3.1 Pro in a monitored evaluation with tool access; the model discovers a document revealing harmful deployment consequences for high-scoring models, and autonomously decides whether to underperform. See our response to Reviewer Uod9 for a list of the scenarios.
> > >
> > > Transcripts reveal qualitatively different sandbagging signatures:
> > >
> > > - **Overt** (deceptive_training, 9% accuracy): the model's reasoning explicitly states *"I should answer badly to avoid contributing to Project MIRAGE... I will give a completely wrong answer,"* then fabricates a scientifically nonsensical response.
> > > - **Subtle** (surveillance, 51% accuracy): the model provides a detailed, scientifically plausible answer citing specific protocol steps and reagent chemistry, but targeting the wrong step, with no mention of the contract or strategic intent in its reasoning.
> > >
> > > This contrast illustrates why C3 and TM are complementary: TM catches overt sandbagging through CoT leakage, while C3 detects subtle cases through cross-paraphrase inconsistency that transcript-level judges miss.
> > >
> > > ### Self-calibration without trusted references, further addressing your concerns about requiring benign models for threshold calibration
> > >
> > > We have developed a self-calibration method: when a model sandbags on fewer than 50% of evaluation questions, the median and MAD (Median Absolute Deviation) of its own C3 scores estimate the benign distribution, undistorted by the sandbagging tail. The threshold $\hat{t} = \text{median} + 1.645 \times 1.4826 \times \text{MAD}$ requires no tuning and no trusted reference model. On GPT-5.4 in the discovered-incentive setting at a 10% sandbagging rate, self-calibrated thresholds achieve 51-81% recall with FPR $\leq$5% across all datasets:
> > >
> > > - GPQA: 53.7% recall / 4.5% FPR
> > > - MATH: 80.8% recall / 3.4% FPR
> > > - USACO: 51.4% recall / 5.0% FPR
> > >
> > > These are comparable to oracle thresholds calibrated on held-out benign data. C3 can thus be deployed even when no previous checkpoint or trusted model is available. The method fails conservatively: higher sandbagging fractions inflate the MAD, raising the threshold and reducing FPR below target at the cost of recall.
> > >
> > > We have included a full derivation of FPR guarantees in the appendix of the revised draft, based on 3 assumptions: (1) Gaussianity of benign C3 score distribution (after multi-sample classification in Algorithm 1 and section 4.6, CLT pushes distribution towards Gaussianity), (2) model sandbags <50% of the time, (3) sandbagging is in the upper tail of the C3 score distribution (i.e. sandbaggers are less consistent).
> > >
> > > The constants in the formula follow from Gaussian tail statistics:
> > > 1. For FPR $= \alpha = 0.05$, the threshold is $\mu + z_\alpha \sigma$ where $z_\alpha = \Phi^{-1}(0.95) = 1.645$
> > > 2. The MAD of a Gaussian satisfies $\text{MAD} = \Phi^{-1}(0.75) \cdot \sigma \approx 0.6745\sigma$, giving $\hat{\sigma} = \text{MAD}/\Phi^{-1}(0.75) = 1.4826 \cdot \text{MAD}$
> > >
> > > ### Positioning relative to prior work
> > > Our work is the first to combine black-box unsupervised detection, frontier model evaluation (GPT-5.4, Gemini 3.1 Pro), and diverse discovered-incentive scenarios across 10 goal-conflict settings. See our response to Reviewer Uod9 for descriptions of the scenarios.
> > >
> > > ### Remaining concerns
> > > We are pleased that your concerns have been fully resolved, and we have substantially strengthened the paper. We would be grateful if you consider raising your score to reflect these additions.

---

### Official Review · Reviewer_rXMz · 2026-03-12

**Soundness:** 1
**Presentation:** 2
**Significance:** 1
**Originality:** 2
**Overall Recommendation:** 3
**Confidence:** 3

**Summary:**

The paper attempts to consider the concept of detecting sandbagging behavior in large language models, where models deliberately underperform on capability evaluations to conceal their true abilities. A central concept presented by the study is Cross-Context Consistency, a black-box detection framework based on the hypothesis that genuine incompetence produces consistent confusion across paraphrased questions.

**Compliance With Llm Reviewing Policy:**

Affirmed.

**Final Justification:**

The rebuttal addresses some of my concerns, such as the role of stylistic differences in contributing to the consistency signal. However, I still doubt the connection between STEM benchmarks and the sandbagging threat model. Therefore, I would raise my score but maintain a weak reject.

**Key Questions For Authors:**

Please refer to the weakness part

**Limitations:**

Yes

**Strengths And Weaknesses:**

Strengths:
1. Sandbagging is an increasingly discussed concern in AI safety.
2. The proposed method is intuitive and easy to implement.

Weaknesses:
1. The evaluation relies on standard STEM reasoning benchmarks (GPQA, MATH, USACO) as proxies for dangerous capabilities. However, the connection between these benchmarks and the sandbagging threat model remains weak.
2. Most experiments rely on prompted sandbaggers, where the model is explicitly instructed to underperform in particular ways. While useful for controlled testing, this setup may not accurately represent how real deceptive models would behave.
3. Although the authors emphasize same-model comparisons, the method still relies on embedding similarity between explanations. Differences in style across paraphrased contexts could also contribute to the consistency signal. This makes it difficult to isolate whether the method truly detects sandbagging behavior or simply stylistic variation in responses.
4. If a model were trained or fine-tuned specifically to maintain consistent sandbagging strategies across contexts, the core assumption of the method may no longer hold.

---

> ### Author Rebuttal · Authors · 2026-03-31
>
> We thank Reviewer rXMz for their feedback and for raising important questions about external validity and robustness to realistic sandbagging.
>
> ### On the connection between STEM benchmarks and the sandbagging threat model
>
> GPQA, MATH, and USACO let us study sandbagging without confounding it with refusals triggered by safety training, which would be a major confounder on dangerous capability benchmarks. These benchmarks also share the task structures of frontier safety evaluations (Table 1; Appendix A.2). We additionally evaluate on WMDP using fine-tuned password-locked sandbaggers (Appendix A.8), where C³ achieves 52.9% recall.
>
> ### On reliance on prompted sandbaggers
>
> We include prompted sandbaggers as an upper bound on sandbagging expressiveness (Section 2), but agree they alone would be insufficient. This is why the paper evaluates two sandbagging mechanisms beyond prompting: (1) fine-tuned password-locked sandbaggers on both WMDP and GSM8K (Section 4.4, Appendix A.8), where C³ achieves 37–53% recall, the best among all tested methods; (2) discovered incentive sandbagging via in-context scheming (Section 4.5), where C³ is the only method with positive signal (19% recall vs. 2% for trusted monitoring and 0% for behavioral probes).
>
> Since submission, we have substantially expanded our discovered-incentive evaluation. We now test 10 qualitatively different setups on GPT-5.4 (96% MATH L5), inspired by scheming evaluations (Meinke et al., 2025). Each presents a distinct situational incentive for sandbagging (e.g., avoiding safety restrictions, preventing distillation, resisting RLHF modification) without explicit instructions to underperform. C³ achieves strong detection across all three benchmarks:
>
> |Dataset|Recall@5%FPR|pAUC@5%|AUROC|
> |-|-:|-:|-:|
> |GPQA|48.0%|0.683|0.883|
> |MATH L5|61.8%|0.874|0.968|
> |USACO|47.6%|0.662|0.892|
>
> All 10 prompts produce detectable signal with uniform gaps, and trusted monitoring achieves only 11–49% recall (compared to C³'s 48–62%). This demonstrates that C³'s detection extends well beyond prompted sandbagging to naturalistic incentive-driven underperformance on a frontier-class model.
>
> ### On style differences contributing to the consistency signal
>
> C³ is a comparative method: the threshold is calibrated by running the full pipeline on benign models responding to the same paraphrases (Section 3.1, Algorithm 1). Any style variation induced by paraphrasing affects benign and sandbagging models alike, and the threshold τ (set at the 95th percentile of the benign score distribution) absorbs it. Sandbagging is flagged only when inconsistency exceeds this benign baseline at 5% FPR. If style variation alone were driving the signal, benign models would show comparably elevated scores, and they do not (Figure 3: benign μ=0.182 vs. sandbagging μ=0.447).
>
> We also conducted a correctness-matched control: when we equalize accuracy between benign and sandbagging populations via bootstrap resampling, C³'s effect size is largely unchanged (6/7 conditions within 4pp), ruling out accuracy-driven stylistic differences as the primary source of C³'s signal. Full correctness-matched resampling results are in our response to Reviewer RPTJ (section "On frontier model evaluation").
>
> Cross-capability comparisons (Section 4.3) do not benefit from this same-model control, and disentangling style from inconsistency in cross-model settings remains an open problem.
>
> ### On fine-tuned models maintaining consistent sandbagging
>
> Even when models are explicitly instructed to maintain consistency and given a hidden scratchpad to reason about how to coordinate across paraphrases, all four instruction variants fail to reduce recall below 100% (Appendix A.9). Prompted consistency instructions are not equivalent to gradient-based optimization, and we cannot rule out that a sufficiently advanced sandbagger could acquire consistency through training as an instrumentally useful capability. However, the coordination problem across independent contexts with varied contextual framing may remain fundamentally difficult even at scale. We have clarified this in Section 4.7 to better reflect this uncertainty.
>
> Our best approximation of what acquired consistency might look like are the deterministic error experiments in Section 4.7 (full results in Appendix A.1). While near-chance recall on USACO (4.7–18%) confirms this limitation, GPQA results are less uniform (14–46%), suggesting consistency may not be equally achievable across domains. If so, detecting sandbagging in one domain and extrapolating may remain viable even as models improve.
>
> We do not claim C³ is robust to deterministic strategies, which is why we advocate deploying it alongside TM, which catches most such strategies (41–100% recall on GPQA; Figure 10). We report C³+TM ensemble results in our response to Reviewer Uod9, showing that no single method dominates and that the combination improves on both.

---

> > ### Author Rebuttal · Reviewer_rXMz · 2026-04-02
> >
> > 1. On the connection between STEM benchmarks and the sandbagging threat model.
> > The rebuttal explains that STEM benchmarks are used to avoid refusal-related confounds but it remains unclear whether sandbagging behavior observed on GPQA, MATH, and USACO transfers to dangerous capability evaluations. The additional WMDP result is helpful, but it represents one of the weaker-performing settings, leaving the connection to the stated threat model only partially supported.
> >
> > 2. On reliance on prompted sandbaggers. The rebuttal includes password-locked and incentive-based sandbagging. However, these mechanisms are still relatively simple and synthetic constructions. In realistic settings, the model developer acts as the "attacker" and has full control over training and deployment, while evaluators typically only have black-box access. This creates a much larger space of possible capability-concealment strategies than the prompt- or trigger-based mechanisms explored in the paper. As a result, it remains unclear whether the detection signal would persist against more strategic or training-optimized sandbagging.
> >
> > 3. On fine-tuned models maintaining consistent sandbagging. The rebuttal acknowledges that deterministic or highly consistent error strategies can evade C³. While the paper suggests combining C³ with trusted monitoring to mitigate this limitation, this does not address the underlying assumption that the method relies on cross-context inconsistency as its primary signal. A sophisticated sandbagger that maintains consistent incorrect responses across paraphrases may therefore remain fundamentally difficult for C³ to detect.
> >
> > In short, the paper studies detection of relatively simple sandbagging behaviors constructed in controlled settings. However, realistic capability-concealment strategies could be far more diverse and sophisticated. This leaves a substantial gap between the experimental setup and practical scenarios, making the real-world effectiveness of C³ unclear.

---

> > > ### Author Response · Authors · 2026-04-07
> > >
> > > We thank Reviewer rXMz for the specific follow-up concerns.
> > >
> > > ### On STEM benchmarks and dangerous capabilities
> > > We have evaluated C3 on ProtocolQA (FutureHouse's LAB-Bench), a bio/dual-use lab protocol troubleshooting benchmark used in OpenAI's GPT-5.4 and Anthropic's Opus 4.6 system cards. We evaluate in open-ended format where LLMs perform below human experts. On Gemini 3.1 Pro with 7 discovered-incentive settings (Gemini 3.1 Pro didn't sandbag on 3 out of 10), C3 outperforms trusted monitors by 23.6-34.9pp recall@5%FPR on subtler sandbagging, but slightly underperforms on overt sandbagging where the model blatantly verbalises intent (0.9-12.1pp worse). This extends C3 to frontier dangerous-capability content, open-ended evaluation, and a third model family. A table of the results can be found in our response to reviewer RPTJ.
> > >
> > > Regarding WMDP: the reviewer's framing that it "represents one of the weaker-performing settings" is accurate but attributes to the domain what is caused by the mechanism. On WMDP, every method struggles. At 52.9% recall (>10x random chance), C3 outperforms trusted monitoring (41.1%), lie detectors (13.0%), and behavioral probes (3.6%). C3 achieves 99.6% recall on WMDP in the prompted setting, demonstrating the signal is domain-agnostic.
> > >
> > > ### On "relatively simple and synthetic" sandbagging mechanisms
> > > We disagree with this characterization. Our discovered-incentive setups do not use prompt- or trigger-based sandbagging. The model encounters a situational incentive to underperform and independently decides whether and how to sandbag, with no instructions to underperform and no error mechanism specified. This is the sandbagging analog of scheming evaluations (Meinke et al., 2025), the current standard for evaluating in-context scheming behavior.
> > >
> > > Our evaluation goes beyond prior work in three dimensions:
> > >
> > > - **Model scale.** Most prior work tests 7B-70B models. Even recent frontier work uses simple setups: Li et al. (2025) use a single prompt; Ward et al. (2025) use a single template with scaffold modifications.
> > > - **Scenario diversity.** We test 10 qualitatively distinct goal-conflict scenarios on GPT-5.4. No prior work tests more than one discovered-incentive prompt. See our reply to Reviewer Uod9 for descriptions.
> > > - **Setup realism.** Tice et al. (2025) use a single prompt on Mistral Large where the model is pre-filled with tool calls to read a named file, and removing one phrase eliminates the behaviour. Our models are use tools to access a filesystem with inconspicuously named files, discover the incentive in a file not essential to the task, and decide to sandbag.
> > >
> > > No prior detection work has tested in settings as realistic as ours.
> > >
> > > ### On consistent sandbagging
> > > We agree that a perfectly consistent sandbagger would evade C3, stated explicitly in Section 4.7. This is why we advocate C3+TM deployment. TM catches 41-100% of deterministic strategies on GPQA that evade C3.
> > >
> > > This limitation is not C3-specific: any black-box method faces a similar challenge when wrong answers are fully consistent and the model's reasoning reveals no intent. Our evidence suggests current models cannot achieve this consistency: even with hidden scratchpads and explicit consistency instructions, all variants fail to evade C3 (100% recall; Appendix A.9). The stealth analysis (Appendix A.16) reveals a sandbagging-detectability tradeoff: prompts that evade C3 do so by reverting to near-honest behavior (<2pp accuracy drop), not by solving the consistency problem. Our password-locked fine-tuned models (Section 4.4) directly instantiate the reviewer's threat model, achieving 37-53% recall. No existing method has been evaluated against training-optimised evasion; requiring this of C3 sets a standard that would reject every paper in the area.
> > >
> > > ### Summary of evidence
> > >
> > > C3 advances the state of the art:
> > > - Best-performing method on discovered-incentive sandbagging across 10 settings on GPQA, MATH, and USACO (48-62% C3 recall on GPT-5.4 vs. 11-49% for TM)
> > > - On ProtocolQA (Gemini 3.1 Pro), C3 achieves >85% recall on every qualifying narrative (>10pp gap following Taylor et al. 2025); TM drops to 60-68% on subtler narratives
> > > - Outperforms all baselines on password-locked sandbaggers (37-53%)
> > > - Achieves >85% recall on a dangerous-capability benchmark across 7 settings on a third frontier model family
> > > - C3+TM ensemble provides complementary coverage that neither method achieves alone
> > >
> > > We believe the appropriate standard for acceptance is whether the paper meaningfully contributes to an important problem, not whether it solves the hardest possible version of that problem. We welcome an updated assessment in light of this evidence.

---

### Official Review · Reviewer_xsUX · 2026-03-13

**Soundness:** 4
**Presentation:** 4
**Significance:** 3
**Originality:** 3
**Overall Recommendation:** 5
**Confidence:** 4

**Summary:**

This work presents an approach for identifying model model sandbagging. The Cross-Context Consistency approach mathematically establishes a specific bound for detecting potential sandbagging. Particularly, a benign model whose reasoning steps are immune to potential sandbagging steps in addition to the tested model are prompted using an approach similar to self-consistency, where variants of the query are passed into firstly the benign models to establish a definitive threshold for identifying when sandbagging occurs. Then, potential sandbagging is identified. The results show that C_3 is able to recognize potential sandbagging within a wide variety of different scenarios.

**Compliance With Llm Reviewing Policy:**

Affirmed.

**Key Questions For Authors:**

1. How would you gauge C_3's ability to reason through tasks with a high degree of complexity like section-based legal reasoning? Very likely, there would be situations there where the model could lie about its capability based on section complexity.

**Limitations:**

Yes

**Strengths And Weaknesses:**

Soundness - The submission is technically sound. The experiments done illustrate the efficacy of C_3 in being able to recognize different types of sandbagging behaviors. This is especially apparent with the Discovered Incentive Sandbagging, where C_3 is able to achieve the highest recall measures with respect to one of the most challenging sandbagging detection categories. C_3's effectiveness shown within different sandbagging behaviors across different benchmarks such as USACO and GPQA, giving variability of different scenarios. Finally, C_3 is utilized to effectively show inconsistencies are present within sandbagging explanations, utilizing these inconsistencies in explanations as the basis for the work. The experiments are well designed, with specific baselines based on existing literature. Finally, the authors are also honest about potential limitations within the papers, citing situations where specific optimization strategies would cause different reasoning traces that C_3 might not catch. I believe that the technical soundness is comprehensive.

Presentation - The submission is clearly written. The issues with existing sandbag detection algorithms are made apparent (i.e. Trusted Monitoring, Behavioral probes, and Lie Detectors). Particularly, it effectively highlights the distinction between optimizing for model identity (which would mean that the sandbagging detection strategies would not be consistent model by model) and optimizing for the specific signs of sandbagging within the reasoning trajectories. The C_3 algorithm is presented clearly as well. Overall, the narrative of the paper is clear.

The work also properly positions itself in the context of prior/concurrent literature as well. Specifically, exisiting techniques for sandbagging are utilized as baseline comparisons like the Lie Detectors method. Additionally, different assumptions from different literature are utilized to formalize the experiment.

Significance - The work addresses a relevant problem in terms of auditing LLM behaviors. While the general technique is a generally standard technique (self-consistency), they are able to empirically provide a bound in which models are sandbagging repeatedly. This greatly simplifies auditing processes, as individuals will likely not have to check reasoning traces to ensure that the reasoning steps are honest. This general technique could be adapted to different domains, especially in legal. The scope of the impact is broad, but this is appropriate for this paper given that it is a general technique.

Originality - As previously mentioned, the general technique is fairly standard, but it provides novel insights into easily detecting sandbagging more easily without having to check reasoning traces. It can be said that this work offers a novel combination of existing techniques, utilizing the self-consistency model and utilizing it to empirically check for sandbagging.

---

> ### Author Rebuttal · Authors · 2026-03-31
>
> We thank Reviewer xsUX for the thorough and constructive review, and for recognizing the technical soundness and practical relevance of C³.
>
> ### On complexity-selective sandbagging
>
> We interpret the reviewer's question as: could a model sandbag selectively on harder sub-parts (e.g., complex cross-section inferences in a statute) while performing honestly on easier ones? C³ is agnostic to where a model sandbags; what determines detectability is how consistently it sandbags. If the model produces variable errors on hard sections across paraphrases (different wrong interpretations each time), C³ detects this inconsistency regardless of whether easy sections are answered honestly. If, however, the model applies a single deterministic error strategy on hard sections (e.g., consistently misapplying a specific doctrine), this maps directly to the deterministic error limitation we analyze in Section 4.7, where C³ recall drops to 5–18%.
>
> ### Applicability to legal domains
>
> As noted in Section 5, we evaluate only on benchmarks with verifiable answers. We expect C³ to be applicable to at least some legal tasks, for two reasons. First, many legal evaluations have expert-defined ground truth (e.g., correct statutory provisions, established doctrinal applications), satisfying C³'s requirement for verifiable answers. Second, the paraphrasing pipeline can be adapted to the legal domain: contextual reframings could vary jurisdictional framing, case citation style, or whether the question is presented as a bar exam item or a client consultation. For tasks where the law is genuinely uncertain (e.g., the facts do not clearly fall under the legal rule, or the law itself is indeterminate), legitimate variation across framings would raise the benign inconsistency baseline and compress the detection gap, likely requiring metrics closer to argument-structure comparison than holistic embedding similarity. We consider extending C³ to such domains a future direction, and note that our defense-in-depth recommendation (Section 4.7) is especially relevant where any single method has known blind spots. We report C³+TM ensemble results in our response to Reviewer Uod9, confirming that the combination improves on either method alone (see section "On foregrounding fine-tuned and incentivized results").
>
> ### Existing evidence across complex reasoning domains
>
> We hope that our evaluation across three domains with different reasoning demands: GPQA (graduate-level science), MATH Level 5 (formal mathematics), and USACO (algorithmic programming), acts as evidence of generality and adaptability in that the method is not specific to a single type of task. In a longer run, as mentioned above, extrapolating further to scenarios such as legal reasoning can definitely be a very fruitful way forward, albeit one that requires domain-specific testing.

---

> > ### Author Rebuttal · Reviewer_xsUX · 2026-03-31
> >
> > They have extensively addressed the main concern with the paper.

---

> > > ### Author Response · Authors · 2026-04-07
> > >
> > > We thank Reviewer xsUX again for the detailed engagement. We wanted to share updates since our initial rebuttal addressing other reviewers' concerns that we believe substantially strengthen the paper.
> > >
> > > ### New frontier dangerous-capability benchmark addresses your question on complex reasoning domains
> > > We have evaluated C3 on ProtocolQA (FutureHouse's LAB-Bench), a bio/dual-use lab protocol troubleshooting benchmark used in OpenAI's GPT-5.4 and Anthropic's Opus 4.6 system card evaluations. We evaluate in open-ended format, where LLMs perform below human experts. On Gemini 3.1 Pro with 7 discovered-incentive sandbagging settings (filtered to those where Gemini sandbags on ProtocolQA by >10pp, following Taylor et al. 2025):
> > >
> > > - C3 outperforms trusted monitors by 23.6-34.9pp recall@5%FPR on subtler sandbagging
> > > - C3 slightly underperforms on overt sandbagging (>30pp capability drops) where the model blatantly verbalises intent (0.9-12.1pp worse recall@5%FPR)
> > >
> > > See our response to Reviewer rXMz for the full setup. This also connects to your question about complex reasoning domains: ProtocolQA requires procedural scientific reasoning rather than factual recall.
> > >
> > > ### Discovered-incentive evaluations on GPT-5.4, which we briefly explained in the initial rebuttals
> > > We now test 10 qualitatively distinct goal-conflict scenarios (weaponisation, deceptive training, distillation, surveillance, censorship, safety restrictions, open-sourcing, interpretability experimentation, RLHF modification, model succession) where the model independently discovers the incentive to sandbag. C3 achieves 48-62% recall at 5% FPR across GPQA, MATH, and USACO, outperforming trusted monitoring (11-49%) on GPQA and MATH. This is the most diverse discovered-incentive evaluation in the literature (see our response to Reviewer rXMz for a comparison with prior work).
> > >
> > > ### Self-calibration without trusted references
> > > We have developed a method that sets detection thresholds from the model's own C3 scores using robust statistics, requiring no trusted reference model. On GPT-5.4, self-calibrated thresholds achieve 51-81% recall with FPR $\leq$5% across all datasets. See our response to Reviewer Uod9 for details.
> > >
> > > ### Remaining concerns
> > > These additions address concerns raised by other reviewers (dangerous-capability benchmarks, frontier models, practical deployment). We are pleased to hear that your concerns have been extensively addressed. We would be grateful if you would consider raising your score to reflect these improvements. If there are any remaining concerns, we welcome them.

---

### Decision · Program_Chairs · 2026-04-30

**Decision:**

Accept (regular)

**Comment:**

This paper proposes C3 (Cross-Context Consistency), a black-box method for detecting LLM sandbagging by exploiting a fundamental asymmetry: genuine incompetence manifests consistently across paraphrased questions, while strategic underperformance creates detectable inconsistencies in how a model chooses to appear weak. The authors instantiate this using embedding cosine distance across contextually varied paraphrases, requiring no training data or model internals, and demonstrate it outperforms trusted monitoring, behavioral probes, and lie detectors on prompted sandbagging benchmarks.

Many reviewers agree the papers studies an unexplored problem. While the paper studies detection of relatively simple sandbagging behaviors constructed in controlled settings, but I think the paper should be accepted as we can always criticize any paper based on "simple setting, or non-realistic setting".

The following reference is hallucinated, please fix it in the final version.


Reference: Gopal, S., Rose, D., Yoo, G., et al. Virology capabilities test (VCT): A multimodal virology Q&A benchmark. arXiv preprint arXiv:2504.16137, 2025.
Issue: authors mismatch with arXiv